# Latency-Constrained Neural Architecture Search Method for Efficient Model Deployment on RISC-V Devices

**Mingxi Xiang, Rui Ding, Haijun Liu and Xichuan Zhou \***

School of Microelectronics and Communication Engineering, Chongqing University, Chongqing 400044, China; mingxi.xiang@cqu.edu.cn (M.X.); dingrui961210@cqu.edu.cn (R.D.); haijun_liu@cqu.edu.cn (H.L.)
\* Correspondence: zxc@cqu.edu.cn

**Abstract:** The rapid development of the RISC-V instruction set architecture (ISA) has garnered significant attention in the realm of deep neural network applications. While hardware-aware neural architecture search (NAS) methods for ARM, X86, and GPUs have been extensively explored, research specifically targeting RISC-V remains limited. In light of this, we propose a latency-constrained NAS (LC-NAS) method specifically designed for RISC-V. This method enables efficient network searches without the requirement of network training. Concretely, in the training-free NAS framework, we introduce an RISC-V latency evaluation module that includes two implementations: a lookup table and a latency predictor based on a deep neural network. To obtain real latency data, we have designed a specialized data collection pipeline for RISC-V devices, which allows for precise end-to-end hardware latency measurements. We validate the effectiveness of our method in the NAS-Bench-201 search space. Experimental results demonstrate that our method can efficiently search for latency-constrained networks for RISC-V devices within seconds while maintaining high accuracy. Additionally, our method can easily integrate with existing training-free NAS approaches.

**Keywords:** RISC-V; neural architecture search; latency-constrained

## 1. Introduction

RISC-V, a free and open instruction set architecture (ISA), has been instrumental in revolutionizing the microprocessor industry. Its widespread adoption can be observed in traditional computing devices, wearable devices, home appliances, and other domains, owing to its cost-effectiveness and superior scalability [1]. As an exceptionally extendable open-source ISA, it is well suited for artificial intelligence applications like image recognition [2], object detection [3], and natural language processing [4].

In recent years, the development of deep neural networks has significantly advanced the field of computer vision. These networks are now prevalent in various embedded device application scenarios, such as virtual reality (VR) systems [5], object detection, tracking [6], and more. The success of computer vision tasks is chiefly credited to the advent of convolutional neural networks (CNNs) [7], renowned for their robust feature extraction, model expressiveness, and generalization capabilities.

However, manually designing competitive deep networks is a laborious task, demanding substantial human effort to identify the optimal network configuration due to the vast design space of neural networks. To tackle this challenge, there has been a surge of research on neural architecture search (NAS) methods, which automate the process of designing high-performance deep neural networks [8]. The proposal of NAS has moved researchers from designing network structures to designing algorithms for searching candidate networks [9,10]. The search process often involves training numerous candidate networks to gather essential information for network evaluation, resulting in significant time consumption during the NAS procedure.

To address this problem, NASWOT [11] and EPE-NAS [12] have proposed their training-free methods for evaluating network accuracy, enabling search high-accuracy

network architecture within seconds. However, these methods solely prioritize accuracy, which could result in networks that fail to meet latency constraints. In latency-sensitive systems, like autonomous driving and drone tracking, strict response time requirements must be satisfied. TAS [13] and DASS [14] have designed methods that combine quantization with NAS, and pruning with NAS, respectively. The goal of these methods is to reduce memory cost and speed up inference. DeepMaker [15] devised a neuro-evolutionary approach that incorporates multiple objectives to search DNN architectures that are nearly optimal in terms of both accuracy and network size. This method aims to strike a balance between achieving high accuracy and keeping the network size as compact as possible. While many studies use the number of floating point operations (FLOPs) to estimate hardware costs [16,17], some research has shown that fewer FLOPs of network architectures may not necessarily lead to higher efficiency [18]. For instance, NasNet-A [10] and MobileNetV1 [19] have similar computational complexities (564M vs. 575M), but MobileNetV1 utilizes hardware-friendly network architectures, resulting in much lower inference latency.

Several studies have suggested using hardware inference latency to evaluate hardware costs. ProxylessNAS [20] introduces a novel hierarchical latency predictor that utilizes a lookup table to estimate the inference latency. However, it should be noted that complex network architectures may not exhibit a linear relation between the latency of individual operators during inference. This is because neural network inference involves parallel computing on hardware [21]. The actual end-to-end inference latency of the network on hardware provides a more accurate reflection of its efficiency. Therefore, BRP-NAS [22] and FastStereoNet [23] have designed latency predictors to estimate end-to-end inference latency on FPGA, desktop CPU, and desktop GPU. While certain studies have measured hardware inference latency on prevalent commercial edge devices such as ARM, FPGA, and Edge GPUs [24,25], no benchmarking has been performed on RISC-V devices. If we employ hardware-aware NAS on devices other than RISC-V, the final searched network may not perform well [24].

The hardware-aware NAS methods mentioned above require network training during the entire search process, resulting in a significant time commitment. Moreover, these approaches do not optimize for RISC-V devices. Therefore, we propose a training-free LC-NAS method specially designed for RISC-V and summarize the contributions of this study as follows:

- We propose the LC-NAS method that integrates latency constraints with the scoring function of existing training-free NAS methods. This approach further reduces the search space of the network and enables the discovery of networks with high accuracy, while also meeting latency constraints. We expedite the exploration of efficient network architectures designed specifically for RISC-V devices, completing the entire process within seconds.
- To incorporate latency constraints, we propose a latency evaluation module. Depending on the exhaustiveness of the search space, this module takes two forms: a lookup table for exhaustive search space and a deep neural network-based (DNN-based) latency predictor for inexhaustible search space. The latency data for the lookup table and predictor are obtained using a latency collection pipeline on RISC-V devices.
- We conduct a comprehensive analysis of the collected latency dataset, which includes examining the correlation between latency on RISC-V devices and other devices within the same networks. We also investigate the correlation between inference latency and network accuracy, among other factors. This work could provide valuable insights for other researchers in developing hardware-aware NAS methods specifically designed for RISC-V devices.

## 2. Related Works

The standard NAS process typically consists of two key components: (1) the search space, which includes various network architectures and hyperparameter ranges, and (2) the search algorithm, which aims to find the optimal architecture within the search space to

maximize the objective function. The search algorithm typically involves a search strategy and an accuracy evaluation module. The search strategy is primarily used to identify candidate networks, while the evaluation module is used to assess their performance. The evaluation module requires training to assess the performance of each candidate architecture, which leads to significant computational costs. Employing traditional NAS methods, the process of designing a suitable neural network architecture for a specific task requires hundreds to thousands of GPU hours.

To eliminate the network training part of the NAS process, training-free NAS is proposed. In training-free NAS, the main objective is to optimize the accuracy evaluation component of the standard NAS, while keeping the other components unchanged. This is achieved by utilizing proxy functions or models for evaluating accuracy. The key principle of training-free NAS is to evaluate network architectures at initialization, obtaining accuracy before network training, to save significant training time for candidate architectures. NASWOT [11], a pioneer in the field of training-free NAS, was the first to propose predicting network performance by leveraging the activation overlap of datapoints in untrained networks. EPE-NAS [12], which serves as our baseline, represents network inputs and outputs using a Jacobian matrix. The Jacobian matrix provides information regarding the relation between the network's output and the input across multiple datapoints. An effective network should possess the capability to differentiate local linear operators for individual data points while generating similar outcomes for similar data points, indicating that they belong to the same class. Therefore, by assessing the correlation among datapoints belonging to the same class, we could determine if an untrained network has the ability to model complex functions.

EPE-NAS [12] assesses the network's capability by modeling complex functions of the covariance matrix for each class within the Jacobian matrix. The covariance matrix of each class is computed to obtain the correlation matrix of the classes. The details can be found in [12]. The scoring function can then be expressed using Equation (1).

$$
s = \begin{cases} \sum_{t=1}^{C} |e_t|, & if\ C \leq \tau \\ \dfrac{\sum_{i=1}^{C} \sum_{j=i+1}^{C} |e_i - e_j|}{\|e\|}, & otherwise \end{cases} \tag{1}
$$

where $e$ is the vector that includes all of the correlation matrix scores, and $e_t$ represents the correlation matrix score for the t-th class. $C$ represents the number of classes in the batch, and $\tau$ is a threshold that determines whether normalization is applied. The purpose of normalization is to reduce class differences. In EPE-NAS [12], $\tau$ is defined as 100, and our work adopts the same value.

In our method, this scoring function will serve as the accuracy evaluation module for evaluating network accuracy.

## 3. Materials and Methods

To obtain customized and efficient networks for RISC-V devices, we propose a latency-constrained NAS approach, as shown in Figure 1. We employ a random search strategy [26] as our search strategy and improve the existing training-free NAS performance evaluation by incorporating a latency evaluation module. Before evaluating the accuracy of the candidate networks, we incorporate a latency evaluation of the network architecture. This module eliminates candidate architectures that do not satisfy the specified latency constraints, and the accuracy evaluation module assesses the remaining networks. To measure the end-to-end inference latency for use in the latency evaluation module, we have developed a versatile latency collection pipeline specifically designed for RISC-V devices.

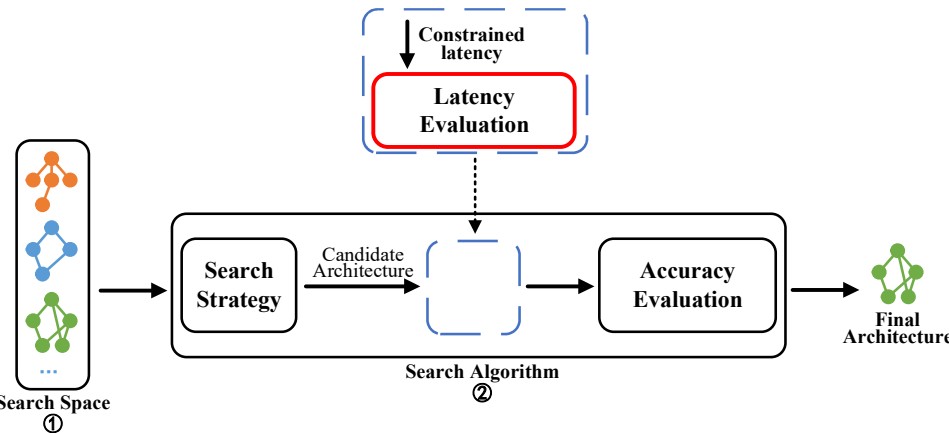

**Figure 1.** Overall framework of our latency-constrained NAS method.

### 3.1. RISC-V Inference Latency Collection Pipeline

To measure the real hardware inference latency of networks, we design a pipeline of latency dataset collection. To address variations in processor speed and other external factors, we repeat the inference process 25 times and record the mean and standard deviation of latency.

Figure 2 presents the latency dataset collection process, which involves the following steps: (1) constructing the model based on the architecture configuration, (2) compiling the neural network model into an executable file compatible with the RISC-V ISA, (3) deploying the model on the target RISC-V hardware board, and (4) measuring the end-to-end inference latency.

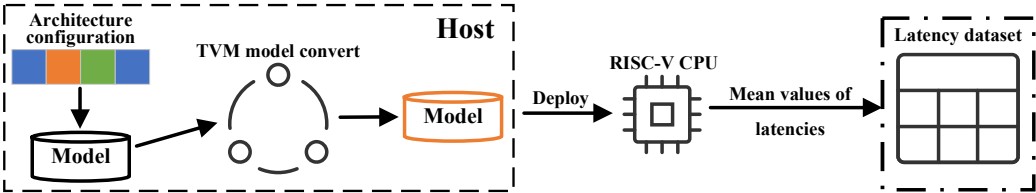

**Figure 2.** Illustration of the RISC-V inference latency collection pipeline.

Specifically, during the compiling step, the TVM framework [27] serves as the deep learning compiler, using the LLVM compiler to generate the executable code. The LLVM compiler is configured to target the riscv64-unknown-linux-gnu platform, with the processor type specified as generic-rv64, and the ABI set as lp64d. Additionally, we enable several RISC-V ISA extensions, including 64-bit operations, integer multiplication and division (M), atomics (A), single-precision floating-point (F), double-precision floating-point (D), and compressed instructions (C).

The host and target board are connected via Ethernet, with the host responsible for compiling the PyTorch model into RISC-V binary executable files. Subsequently, these executable files are transmitted to the target board using the remote procedure call (RPC) protocol for multiple runs, and the mean and standard deviation of the model's inference latency are returned via RPC.

### 3.2. LC-NAS

By utilizing the latency collection pipeline, we gather end-to-end inference latency data for RISC-V devices. These data are subsequently used in the latency evaluation module of LC-NAS. Algorithm 1 presents the pseudocode for the proposed method, which integrates the latency evaluation module with the training-free NAS method. Depending on the size of the search space, an appropriate Latency_eva function can be selected between the lookup table and the DNN-based predictor. Acc_eva function is the accuracy evaluation module. From the search space, $N$ candidate architectures are randomly selected, and latency

evaluation is performed on these untrained networks. Subsequently, the networks that meet the latency constraint $L_c$ are evaluated by the accuracy evaluation module, and the network *Arch* with the highest score is selected as the final network.

---

**Algorithm 1** LC-NAS

---

**Input:** predefined search space, latency evaluation module, accuracy evaluation module, latency constraint $L_c$, number of sampled architectures $N$.
**Output:** searched neural architecture: *Arch*.
 1: initialize *score* = zeros(1, *N*)
 2: *A* = random(*N*, *search_space*)
 3: **for** *i* = 0 to *N* **do**
 4:     *pred* = Latency_eva(*A*[*i*])
 5:     **if** *pred* $\leq L_c$ **then**
 6:         *score*[*i*] = Acc_eva(*A*[*i*])
 7:     **end if**
 8: **end for**
 9: *idx* = argmax(*score*)
10: **return**  *Arch* = *A*[*idx*]

---

The latency evaluation module serves to provide network latency information during the latency-constrained NAS process. We classify it into two types based on whether the search space can be exhaustively enumerated.

- The search space can be exhaustively enumerated. We employ a lookup table to evaluate the latency.
- The search space cannot be exhaustively enumerated. Instead, we randomly gather latency data for a subset of the search space and use this data to train a DNN-based latency predictor.

### 3.2.1. Lookup Table for Latency

In cases where search spaces can be exhaustively enumerated, we can collect inference latency for all candidate networks in the search space by employing the previously mentioned method to collect latency data. This approach is particularly effective for exhaustive search spaces because obtaining the actual end-to-end inference latency directly on hardware provides a more realistic representation of real-world scenarios. This enables the identification of the most efficient neural network for the target hardware. By creating a lookup table from the latency data, we can quickly retrieve the network information using the network architecture name as soon as a candidate network is found. This allows for a latency evaluation to be completed in milliseconds.

### 3.2.2. DNN-Based Latency Predictor

In cases where search spaces cannot be exhaustively enumerated, it is impossible to acquire the inference latency for all candidate networks. Furthermore, as the size of the search space increases, the time required to search the table also increases. When the search space reaches a certain threshold, the time spent searching the table becomes long.

Consequently, we propose a DNN-based predictor to estimate end-to-end inference latency. This predictor takes into account factors such as the architecture and computational complexity of the candidate networks. Subsequently, we only need latency information for a subset of the search space to predict the latency for all candidate networks. The predictor consists of an input layer, a one-hot encoding layer, a concatenation layer, and a fully connected layer.

(1) Architecture Encoding: Prior to prediction, the network architecture is encoded. It is evident that network inference latency is directly related to the network architecture. The majority of candidate networks in the search space consist of predefined architec-

ture and partially variable blocks. Therefore, encoding specific modules in the network architecture as vectors is used as network architecture features.

Taking NAS-Bench-201 [28] as an example, each network architecture in NAS-Bench-201 [28] consists of a predefined backbone network with the searched cells stacked 5 times at 3 preset positions, respectively. Therefore, NAS can be considered as a search process for an efficient cell architecture. Each cell can be represented as a directed acyclic graph (DAG) with dense connections, consisting of 4 nodes and 6 edges. The connection options of each node include 5 representative operations: (1) zeroize, (2) skip connection, (3) $1 \times 1$ convolution, (4) $3 \times 3$ convolution, and (5) $3 \times 3$ average pooling layer. The convolution within this connection set represents a condensed sequence of operations, including ReLU, convolution, and batch normalization. The zeroize operation is used to remove the corresponding edge. A detailed explanation of the cell architecture encoding scheme is presented in Figure 3. Utilizing a predefined one-hot encoding table, we encode the connection options of each node in the cell, resulting in six one-hot codes. These codes are then concatenated into a $1 \times 30$ vector, which serves as a partial input to the predictor.

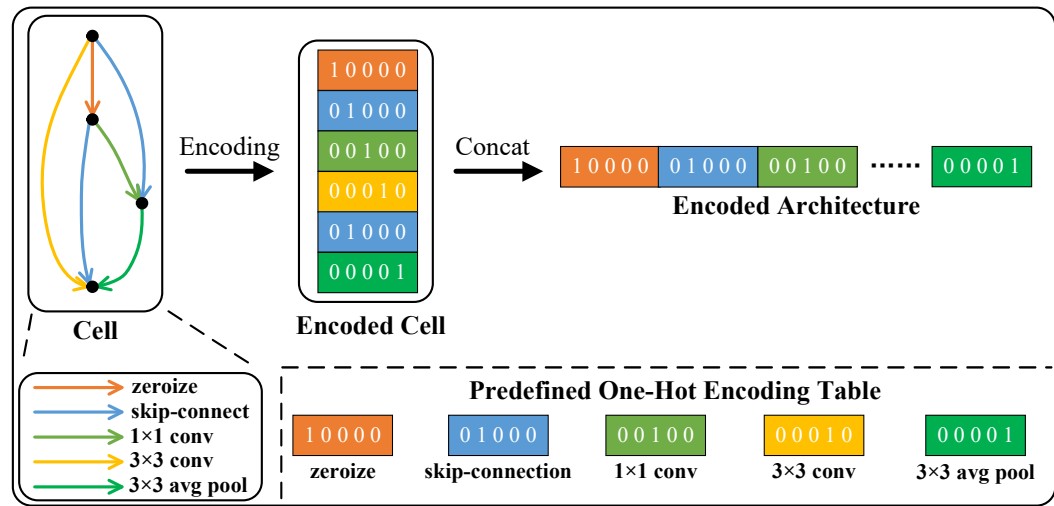

**Figure 3.** A detailed explanation of NAS-Bench-201 cell architecture encoding scheme.

(2) DNN-based Prediction Network: Inference latency is commonly influenced by network architecture and computational complexity. The inference latency of the network should be related to the input size, FLOPs, and architecture. Therefore, the predictor takes 3 types of input: the input size, FLOPs, and variable blocks. The prediction network architecture is illustrated in Figure 4. For non-numeric features like connection options, we employ the one-hot encoding scheme as described earlier to encode the network architecture. For numeric features such as input size and FLOPs, we directly concatenate them with the one-hot encoded architecture. In the NAS-Bench-201 [28], this results in a final $1 \times 32$ vector feature representation. Then, the vector serves as the input for three fully connected layers, each with 32, 64, and 1 neurons, respectively. The ReLU activation function is utilized. It has a total of 3233 trainable parameters.

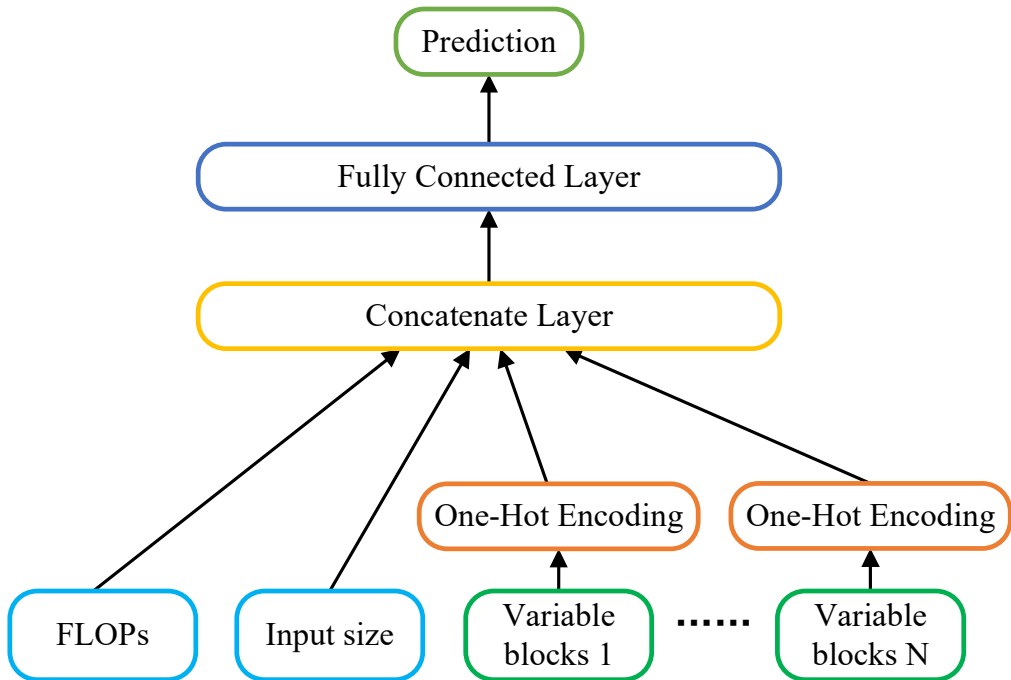

**Figure 4.** Illustration of the DNN-based latency prediction network architecture.

## 4. Experiments and Results

To validate the effectiveness of our proposed method, we conduct experiments within the NAS-Bench-201 [28] search space. We select the NezhaD1 (https://d1.docs.aw-ol.com/d1_dev/, accessed on 27 April 2021) as the RISC-V hardware platform for measuring inference latency. The NezhaD1 is an embedded device equipped with a single-core 1 GHz Xuantie C906 processor and 1 GB of memory. It is manufactured by All Winner Technology Corporation in Zhuhai, China. All experiments are conducted on the Ubuntu20.04 platform using PyTorch 1.13.1, Python 3.7.16, and CUDA 12.0. The CPU utilized in the experiments is the CPU Intel (R) Core (TM) i7-6950X CPU @ 3.00 GHz, while the GPU employed is a single NVIDIA TITAN X Pascal.

### 4.1. Analyzing on RISC-V Latency Dataset

We analyze and visualize the inference latency and corresponding accuracy data for all candidate architectures on CIFAR-10 and ImageNet16-120 in the NAS-Bench-201 [28] search space.

(1) Correlation between Different Devices: We conduct a visualization to observe the inference latency correlation of the same network architecture on different devices. For this purpose, we analyze the inference latency on the other 5 devices obtained from HW-NAS-Bench [24] and compare them with the latency on the RISC-V device. Figure 5 illustrates the Kendall correlation coefficients of inference latency between the RISC-V device and other devices with different system architectures. The Kendall correlation coefficient is a rank correlation coefficient used to assess the strength of the monotonic relation between two ordered variables. Its values range from −1 to 1, where a larger absolute value indicates a stronger monotonic correlation and a value of 0 suggests no correlation.

These figures suggest significant variations in inference latency for the same network architectures when executed on different devices. These variations arise from disparities in the underlying hardware micro-architectures and the available hardware resources. Thus, to obtain efficient neural networks on RISC-V devices, device-specific NAS methods should be employed.

(2) Relation between Latency and Other Architecture Information: To assess the necessity of LC-NAS, we conduct an analysis of the accuracy, FLOPs, and latency of all architectures on the CIFAR-10 and ImageNet16-120 within the NAS-Bench-201 [28] search space.

- In Figure 6a,b, it is evident that an increase in network inference latency does not necessarily result in improved accuracy. Notably, network architectures with significantly reduced inference latency can achieve similar accuracy as more complex ones. Hence, the adoption of LC-NAS becomes crucial.
- In Figure 6c,d, we observe a weak positive relation between FLOPs and latency, so it is inaccurate to use FLOPs alone as prediction information. However, it could be valuable if we combine it with network architecture information as an input for latency prediction.

In order to introduce latency constraints into the NAS process, we need to use a lookup table or a DNN-based latency predictor. The lookup table can be built directly using the latency dataset, while the latency predictor requires training.

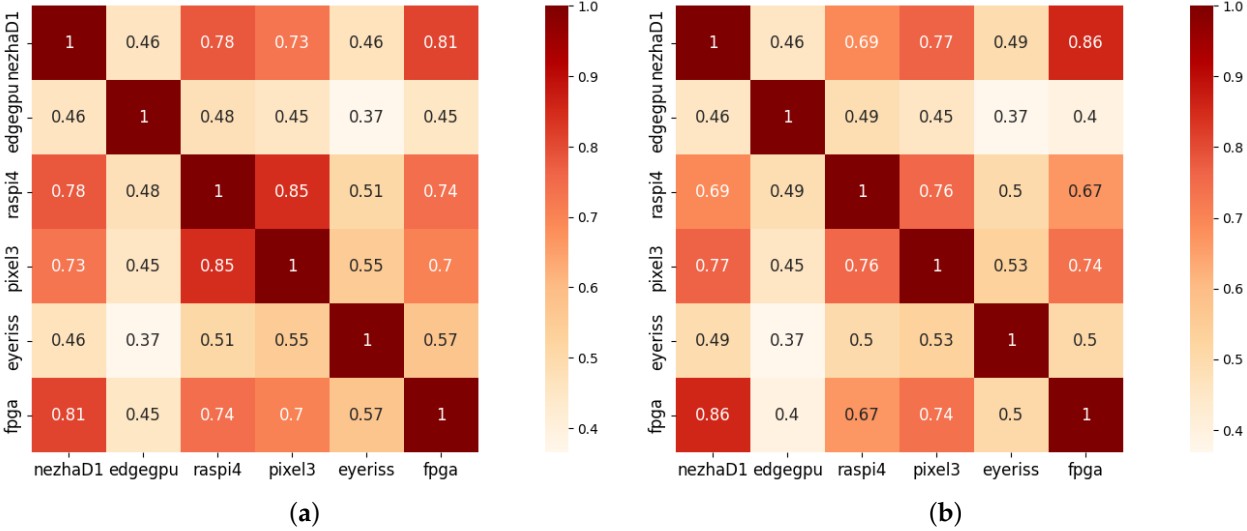

(**a**)　　　　　　　　　　　　　　　　　　　　　　　　　(**b**)

**Figure 5.** The Kendall rank correlation coefficient between the inference latency on RISC-V and other 5 devices on CIFAR-10 and ImageNet16-120. (**a**) Correlation coefficient between different devices on CIFAR-10; (**b**) Correlation coefficient between different devices on ImageNet16-120.

### 4.2. DNN-Based Predictor Training

(1) Training Settings: To train the predictor, we employ the RISC-V inference latency collection pipeline to obtain a latency dataset. 70% of the dataset is used as the training set, while the remaining is used for testing. The widely used Adam optimizer with a learning rate of $1 \times 10^{-3}$ is utilized. The mean squared error loss serves as the loss function, and the batch size is set to 1024.

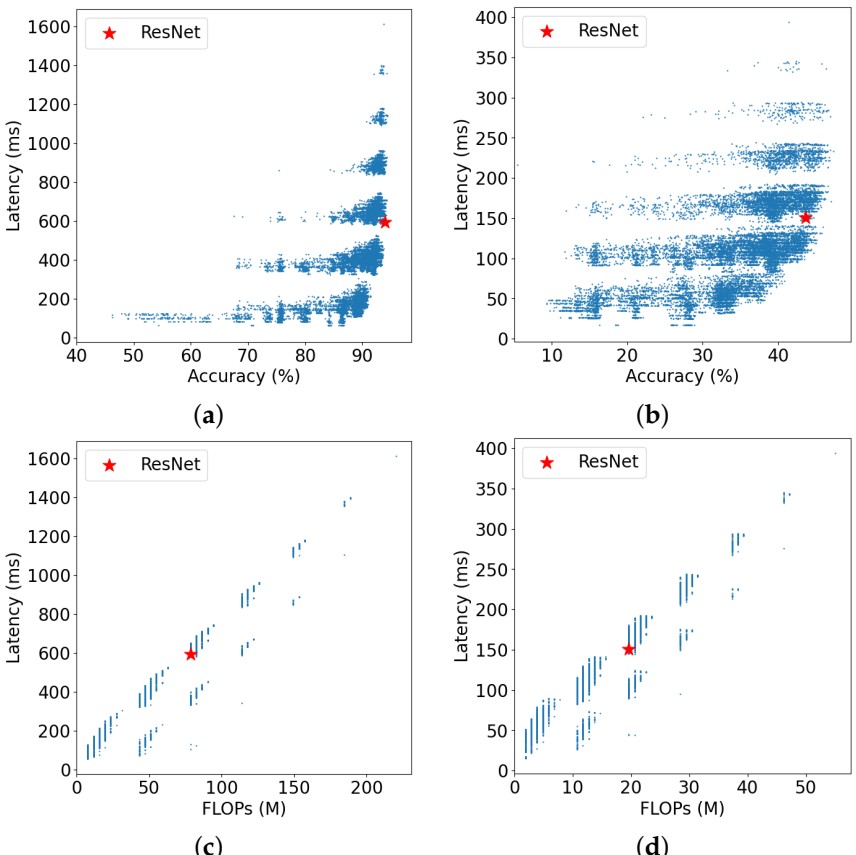

**Figure 6.** Relation between latency and other information of all architectures on CIFAR-10 and ImageNet16-120. (**a**) Relation between latency and accuracy on CIFAR-10; (**b**) Relation between latency and accuracy on ImageNet16-120; (**c**) Relation between latency and FLOPs on CIFAR-10; (**d**) Relation between latency and FLOPs on ImageNet16-120. Where "ResNet" denotes candidate networks with unit architectures identical to the residual network architectures designed by [29].

(2) Comparison with Other Methods: We conduct comparative experiments among commonly used regression prediction models, including Support Vector Regression (SVR), Linear Regression (LR), Elastic-Net, Gradient Boosting Decision Tree (GBDT) [30], BRP-NAS [22], and our latency predictor. The results are illustrated in Table 1. All methods except BRP-NAS [22] use the same input, which is a $1 \times 32$ vector. Table 1 lists the results of all methods evaluated on the latency dataset, with root mean squared error (RMSE), mean absolute error (MAE), coefficient of determination ($R^2$), and Pearson correlation coefficient (Corrcoef). The RMSE metric is more responsive to larger error values, whereas the MAE is more sensitive to smaller ones. Our method exhibits substantial improvement over other approaches concerning the RMSE metric, demonstrating that our prediction method yields smaller errors and the predicted latency is closer to the true values. To ensure fairness, we conduct an experiment in the same evaluation metric as BRP-NAS, i.e., the percentage of models whose predicted latency falls within the specified error bound of the measured latency. The results are illustrated in Table 2.

**Table 1.** Comparison with other prediction methods.

| Method | RMSE ↓ | MAE ↓ | $R^2$ ↑ | Corrcoef ↑ |
|---|---|---|---|---|
| SVR | 28.63 | 7.30 | 0.9868 | 0.9934 |
| LR | 9.91 | 5.08 | 0.9984 | 0.9992 |
| Elastic-Net | 11.94 | 4.88 | 0.9977 | 0.9988 |
| GBDT | 6.77 | 2.28 | 0.9992 | 0.9996 |
| BRP-NAS (CIFAR-10) | 93.42 | 38.60 | 0.8539 | 0.9329 |
| BRP-NAS (ImageNet16) | 22.87 | 10.83 | 0.8495 | 0.9289 |
| Ours | 2.89 | 2.11 | 0.9998 | 0.9999 |

Arrows indicate the relationship between values and performance. ↓ means that lower values present better performance, ↑ means the opposite.

**Table 2.** Comparison with BRP-NAS [22] predictor method in error bound.

| Error Bound | Our Predictor (%) | | BRP-NAS Predictor (%) | |
|---|---|---|---|---|
| | Cifar10 | ImageNet16-120 | Cifar10 | ImageNet16-120 |
| ±1% | 76.18 | 33.61 | 32.92 | 23.64 |
| ±5% | 97.85 | 89.34 | 66.89 | 59.66 |
| ±10% | 99.46 | 96.74 | 76.56 | 73.17 |
| ±20% | 99.87 | 99.28 | 83.52 | 82.75 |

In Table 1, it can be observed that our latency predictor outperforms the other methods in all evaluation metrics. Compared to the best method GBDT [30], the RMSE of the predicted latency of our method is reduced by 57% in the case of a similar MAE. Of these, the RMSE and MAE of BRP-NAS [22] on CIFAR-10 are significantly higher than our method and other prediction methods. Our method is associated with a 96% reduction in RMSE and a 94% reduction in MAE metrics when compared to BRP-NAS [22] on CIFAR-10, respectively. In Table 2, when the error bound is ±10%, the accuracy of our predictor is 20% higher than that of BRP-NAS [22]. The results are analyzed as follows.

- Compared with SVR, LR, Elastic-Net, and GBDT [30], our predictor with deep neural networks enhances its nonlinear learning capabilities, leading to better fitting of latency data.
- Compared with BRP-NAS [22], we use the different input features. We consider not only the neural network architecture but also the FLOPs and input size. However, BRP-NAS only uses architecture information. In Figure 6c,d, it can be observed that there is a relation between FLOPs and latency. It could be valuable to use it as one of the inputs.
- Our method is more generalizable; we can use one predictor for the inference latency of CIFAR-10 and ImageNet16-120. We can use the real latency of both CIFAR-10 and ImageNet16-120 for training, so the predictor can learn more latency information. However, BRP-NAS [22] is trained using information from a single latency dataset; it cannot predict both CIFAR-10 and ImageNet16-120 at the same time.

### 4.3. Training-Free NAS with Latency Constraint

As an example of our proposed LC-NAS method, we use EPE-NAS [12] as our baseline to search within the NAS-Bench-201 [28] search space on CIFAR-10, and ImageNet16-120, respectively. The accuracy values of each neural architecture are directly obtained from NAS-Bench-201 [28]. The image size of CIFAR-10 is $3 \times 32 \times 32$, and the image size of ImageNet16-120 is $3 \times 16 \times 16$. In our experiments, we run the NAS algorithm 20 times to obtain 20 final architectures and then calculate the mean and standard deviation of accuracy and latency.

(1) Search with Different Latency Constraint: The experimental results of baseline and our method are listed in Tables 3 and 4. The parameter $N$ represents the number of ran-

domly sampled network architectures, which is 1000. $L_c$ is the latency constraint measured in milliseconds (ms). Through Tables 3 and 4, we obtain the following conclusions.

- Referring to Tables 3 and 4, we observe that our method demonstrates a faster search speed compared to EPE-NAS [12] and NASWOT [11], as certain candidate networks are eliminated due to the imposed latency constraints. Only networks that meet the latency constraints will be considered for evaluation in the accuracy evaluation module. When the constraints become less strict, the accuracy of the searched networks gradually converges toward the baseline.

- In Table 3, when the latency constraint is set to 500 ms, we achieve higher-accuracy networks using 58% of the search time of NASWOT [11] with the predictor, and similar accuracy networks using only 43% of the search time of NASWOT [11] with the lookup table. Compared to the baseline, the search time is reduced to 90% and 68% of the original, respectively, and the inference latency is also decreased.

- Networks that satisfy the latency constraints can be obtained using both the lookup table and the latency predictor, and both approaches can speed up the NAS process. In this experiment, the search space chosen is NAS-Bench-201 [28], which has a search space size of 15,625. The lookup table is faster than the predictor. Because the lookup table stores the latency of all network architectures in advance, the latency can be obtained by simply performing a traversal lookup. Therefore, the search time of the lookup table is related to the size of search space. As the search space increases, the search time for the lookup table also increases, and may even result in failure to build the lookup table. The time taken by the predictor does not vary with the size of the search space; it remains constant. Therefore, for large search spaces, the predictor could be a better choice.

- Figure 7a illustrates the distribution of all candidate network latency for CIFAR-10, where a large number of neural architectures' latency are concentrated between 300 ms and 450 ms. As a consequence, in Table 3, when $L_c$ is loosened from 300 ms to 500 ms, there is a notable increase in the search time. Figure 7b shows the distribution of latency for ImageNet16-120, and most of the latency is distributed between 100 ms and 150 ms intervals. Therefore, when $L_c$ is set from 100 ms to 150 ms in Table 4, the search time increases by a large amount.

(2) Search with Different Sampling Number: Table 5 reports the results of different sampling number $N$ on CIFAR-10 dataset, where $L_c$ is set 500 ms. Our method can search networks that meet the specified latency constraints for various sampling numbers. With a decrease in the sampling number, the search time also decreases, while the accuracy of the identified networks remains relatively constant. When the sampling number is set from 10 to 1000, the search time.

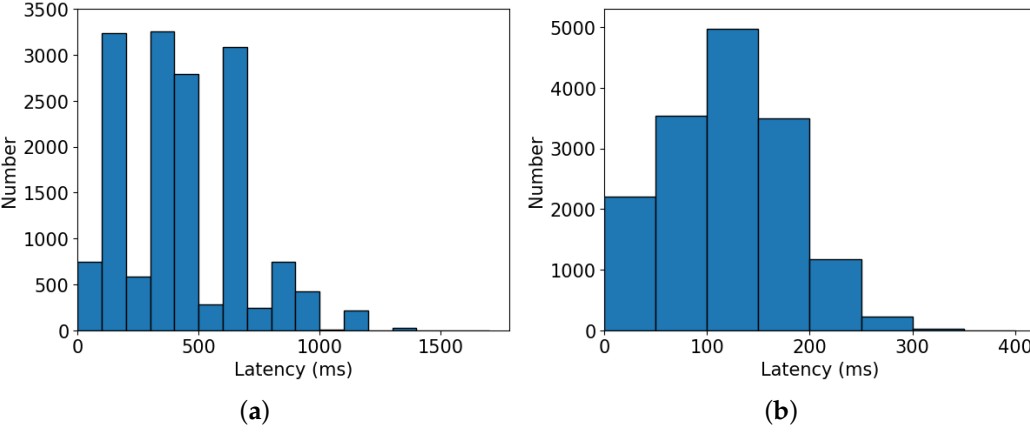

(a)　　　　　　　　　　　　　　　　(b)

**Figure 7.** Distribution of inference latency on (**a**) CIFAR-10 and (**b**) ImageNet16-120 datasets, respectively.

**Table 3.** Performance comparisons between LC-NAS and baseline on CIFAR-10. The mean $\pm$ std search time, accuracy, and real latency are reported.

| Method | Latency Constraint (ms) | Search Time (s) | CIFAR-10 | |
| --- | --- | --- | --- | --- |
| | | | Acc (%) | Latency (ms) |
| EPE-NAS (baseline) | None | 223.69 $\pm$ 1.57 | 91.38 $\pm$ 1.85 | 461.60 $\pm$ 291.77 |
| NASWOT | None | 347.09 $\pm$ 5.17 | 91.22 $\pm$ 1.93 | 328.54 $\pm$ 133.23 |
| With Latency Predictor | | | | |
| Ours | 100 | 54.40 $\pm$ 0.90 | 86.69 $\pm$ 1.21 | 77.47 $\pm$ 11.95 |
| Ours | 300 | 109.55 $\pm$ 3.22 | 88.24 $\pm$ 0.97 | 173.44 $\pm$ 46.91 |
| Ours | 500 | 202.15 $\pm$ 2.64 | 91.26 $\pm$ 1.51 | 374.60 $\pm$ 91.67 |
| With Lookup Table | | | | |
| Ours | 100 | 33.70 $\pm$ 0.73 | 86.43 $\pm$ 0.20 | 76.06 $\pm$ 13.18 |
| Ours | 300 | 77.04 $\pm$ 2.45 | 89.18 $\pm$ 1.14 | 172.66 $\pm$ 43.67 |
| Ours | 500 | 152.67 $\pm$ 2.21 | 91.19 $\pm$ 1.47 | 340.23 $\pm$ 131.83 |

**Table 4.** Performance comparisons between LC-NAS and baseline on ImageNet16-120.

| Method | Latency Constraint (ms) | Search Time (s) | ImageNet16-120 | |
| --- | --- | --- | --- | --- |
| | | | Acc (%) | Latency (ms) |
| EPE-NAS (baseline) | None | 176.08 $\pm$ 0.78 | 40.29 $\pm$ 3.81 | 134.13 $\pm$ 62.63 |
| NASWOT | None | 273.24 $\pm$ 2.35 | 40.19 $\pm$ 3.38 | 107.82 $\pm$ 46.92 |
| With Latency Predictor | | | | |
| Ours | 50 | 68.41 $\pm$ 1.37 | 31.16 $\pm$ 3.75 | 45.15 $\pm$ 4.56 |
| Ours | 100 | 103.23 $\pm$ 1.89 | 32.72 $\pm$ 6.93 | 75.38 $\pm$ 21.52 |
| Ours | 150 | 159.14 $\pm$ 1.99 | 39.31 $\pm$ 4.63 | 111.55 $\pm$ 24.84 |
| With Lookup Table | | | | |
| Ours | 50 | 46.85 $\pm$ 1.47 | 31.57 $\pm$ 2.37 | 37.70 $\pm$ 8.89 |
| Ours | 100 | 76.56 $\pm$ 1.45 | 33.41 $\pm$ 3.27 | 62.73 $\pm$ 20.04 |
| Ours | 150 | 122.42 $\pm$ 1.78 | 39.77 $\pm$ 3.63 | 106.25 $\pm$ 26.76 |

**Table 5.** Different sampling number *N* of LC-NAS on CIFAR-10.

| N | Search Time (s) | Acc (%) | Latency (ms) |
| --- | --- | --- | --- |
| With Latency Predictor | | | |
| 10 | 2.03 $\pm$ 0.34 | 90.08 $\pm$ 3.19 | 318.12 $\pm$ 110.95 |
| 100 | 20.49 $\pm$ 0.99 | 91.22 $\pm$ 1.55 | 373.58 $\pm$ 78.33 |
| 500 | 101.79 $\pm$ 2.31 | 91.50 $\pm$ 0.68 | 412.77 $\pm$ 40.96 |
| 1000 | 202.15 $\pm$ 2.64 | 91.26 $\pm$ 1.51 | 374.60 $\pm$ 91.67 |
| With Lookup Table | | | |
| 10 | 1.58 $\pm$ 0.26 | 90.82 $\pm$ 1.99 | 296.82 $\pm$ 130.57 |
| 100 | 15.41 $\pm$ 0.76 | 90.41 $\pm$ 2.05 | 298.67 $\pm$ 121.72 |
| 500 | 76.25 $\pm$ 1.74 | 90.85 $\pm$ 2.06 | 319.15 $\pm$ 126.07 |
| 1000 | 153.23 $\pm$ 2.11 | 91.19 $\pm$ 1.47 | 340.23 $\pm$ 131.83 |

## 5. Conclusions

The current training-free NAS methods fail to consider the inference latency of devices and instead focus exclusively on network accuracy. Moreover, the hardware-aware NAS is not optimized for RISC-V devices. Therefore, this paper proposes LC-NAS method specifically designed for RISC-V devices, which can be seamlessly integrated with existing

training-free methods. The method searches for high-accuracy networks while ensuring compliance with latency constraints. The introduction of latency constraints helps reduce the search space for networks, resulting in the acceleration of the NAS process. The visualization of the experiment demonstrates that the architectures of efficient networks vary significantly across different hardware devices. Hence, it is crucial to perform network searches specifically tailored for RISC-V devices.

**Author Contributions:** Conceptualization, X.Z.; Methodology, M.X. and R.D.; Software, M.X; Validation, H.L.; Investigation, X.Z.; Resources, H.L.; Data curation, R.D.; Writing—original draft, H.L. and M.X.; Writing—review & editing, X.Z., H.L., R.D. and M.X.; Visualization, M.X.; Supervision, X.Z. and H.L.; Project administration X.Z.; Funding acquisition, X.Z. All authors have read and agreed to the published version of the manuscript.

**Funding:** This paper was supported in part by the National Natural Science Foundation of China under Grants U2133211, 62001063, and 61971072; and in part by the Fundamental Research Funds for the Central Universities under Grant 2023CDJXY-037.

**Data Availability Statement:** The latency dataset and code are openly available in GitHub at https://github.com/Sunnycee99/LC-NAS, accessed on 28 January 2024.

**Conflicts of Interest:** The authors declare no conflicts of interest.

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
