# Peer review of "Latency-Constrained Neural Architecture Search Method for Efficient Model Deployment on RISC-V Devices"

_electronics, doi:10.3390/electronics13040692_

Round 1

Reviewer 1 Report

Comments and Suggestions for Authors

Latency-Constrained Neural Architecture Search Method for Efficient Model Deployment on RISC-V Devices

Summary

·         The paper introduces a latency-constrained Neural Architecture Search (LC-NAS) method tailored specifically for the RISC-V instruction set architecture, addressing a research gap in hardware-aware methods for RISC-V in the context of deep neural network applications. The proposed LC-NAS method employs a training-free framework with a RISC-V latency evaluation module, featuring a lookup table and a deep neural network-based latency predictor.

Score: Major Revision

Review

·         Writing:

o    The paper is well-written and easy to follow.

·         Literature Review:

o    Several related efforts in literature related to HW-aware NAS and latency predictors for designing HW-aware model have been missing from the related work section. I recommend authors adding the following related papers:

§  Loni, Mohammad, et al. "Faststereonet: A fast neural architecture search for improving the inference of disparity estimation on resource-limited platforms." IEEE Transactions on Systems, Man, and Cybernetics: Systems 52.8 (2021): 5222-5234.

§  Loni, Mohammad, et al. "DeepMaker: A multi-objective optimization framework for deep neural networks in embedded systems." Microprocessors and Microsystems 73 (2020): 102989.

§  Dudziak, Lukasz, et al. "Brp-nas: Prediction-based nas using gcns." Advances in Neural Information Processing Systems 33 (2020): 10480-10490.

§  Loni, Mohammad, et al. "Tas: ternarized neural architecture search for resource-constrained edge devices." 2022 Design, Automation & Test in Europe Conference & Exhibition (DATE). IEEE, 2022.

§  Loni, Mohammad, et al. "Learning Activation Functions for Sparse Neural Networks." arXiv preprint arXiv:2305.10964 (2023).

§  Mousavi, Hamid, et al. "DASS: Differentiable Architecture Search for Sparse Neural Networks." ACM Transactions on Embedded Computing Systems 22.5s (2023): 1-21.

·         Novelty:

o    The idea of the prediction method is very interesting.

·         Reproducibility of Results:

o    IMPORTANT. The reproducibility of the proposed method is under question, e.g., the codes are not available and NAS results have been reported with one seed. To guarantee the reproducibility of NAS’s results, check the following paper:

§  Lindauer, Marius, and Frank Hutter. "Best practices for scientific research on neural architecture search." The Journal of Machine Learning Research 21.1 (2020): 9820-9837.

·         Results:

o    Please compare the proposed solution with other latency predictors such as brp-NAS:

§  Dudziak, Lukasz, et al. "Brp-nas: Prediction-based nas using gcns." Advances in Neural Information Processing Systems 33 (2020): 10480-10490.

Comments on the Quality of English Language

No comments

Reviewer 2 Report

Comments and Suggestions for Authors

This paper does a research in the waste domain of neural architecture search, where issues regarding the accuracy and complexity of the networks appear, the designs are usually proper for a specific task, they have to be hardware aware and encounter constraints regarding the memory footprint, interference latency or energy consumption. In this paper a Nas method was proposed specific for RISC-V set architecture, designed observing the latency data and evaluation.

The documentation part provides a pretty clear view of the study in this domain, presenting, besides the subjects mentioned above, also a very new and interesting aspect and algorithm regarding the use of neural networks without training and based on the initial data.

The contributions of the study are clarely stated in the beginning and the end of the work, block diagrams suggestively describe the research process, method and model; the chapters deal with presenting the technical elements that compose the test environment, the lookup table and the latency predictor. Having these aspects as premises, the experimental results are commented, regarding the use of the same network architecture on different devices, the specialized data collection pipeline,  the latency in respect to the architecture information, the NAS training-free with the constraints imposed by latency, all summarized in the end in Tables 1 to 4, where the strong points and the weaknesses of the method are highlighted. The method validation is also shown in the conclusion comments.

 Therefore, the research work leads to find efficiently latency constrained networks for RISC-V, keeping a high accuracy, where the focus is. The latency constraints help reducing the search space, but it is highlighted the fact that the architectures performances depend on the hardware devices, therefore it is indicated that the researches are performed specifically on devices, in this case on RISC-V.

Author Response

Thank you very much for taking the time to review our manuscript entitled “Latency-Constrained Neural Architecture Search Method for Efficient Model Deployment on RISC-V Devices”. We have carefully considered the comments and improved the method section for adequate description, which can be found in our modified manuscript.

Reviewer 3 Report

Comments and Suggestions for Authors

Comments on the Quality of English Language

Round 2

Reviewer 1 Report

Comments and Suggestions for Authors

Thanks for applying my comments!

Reviewer 3 Report

Comments and Suggestions for Authors

Comments on the Quality of English Language